# Combined analysis of mRNA and miRNA reveals the mechanism of pacific white shrimp (*Litopenaeus vannamei*) under acute alkalinity stress

**Xiang Shi, Ruiqi Zhang** **\*, Zhe Liu, Jun Sun, Lanlan Li, Guiyan Zhao, Junhao Lu**

College of Animal Science & Technology, Gansu Agricultural University, Lanzhou, Gansu Province, China

* mading000000@126.com

**Data Availability Statement:** The datasets presented in this study can be found in the NCBI repository (https://www.ncbi.nlm.nih.gov/sra)

## Abstract

The pacific white shrimp (*Litopenaeus vannamei*) is now a more common aquaculture species in saline-alkali waters, while alkalinity stress is considered to be one of the stressors for shrimp. Thus, an understanding of the molecular response to alkalinity stress is critical for advancing the sustainability of culture in pacific white shrimp. In this study, we aimed to explore the response mechanism to acute high-alkaline stress by RNA-seq at low-alkaline (50 mg/L) and high-alkaline (350 mg/L). We identified 215 differentially expressed mRNAs (DEGs) and 35 differentially expressed miRNAs (DEMs), of which 180 DEGs and 28 DEMs were up-regulated, 35 DEGs and 7 DEMs were down-regulated, respectively. The DEGs were enriched in several pathways, including carbohydrate digestion and absorption, pancreatic secretion, starch and sucrose metabolism, antigen processing and presentation and glutathione metabolism. The DEMs involved in lysosome and ion transport related pathways were significantly up-regulated. We also achieved 42 DEGs, which were targeted by DEMs. miRNA-mRNA regulatory network was constructed by integrated analysis of miRNA-mRNA data. We detected several genes and miRNAs which were identified as candidate regulators of alkalinity stress, and expression patterns of key genes related to alkalinity stress in pacific white shrimp. Among these genes, the expression levels of most key genes enriched in ion regulation, digestion and immunity were increased, and the expression levels of genes enriched in metabolism were down-regulated. This research indicated that the homeostatic regulation and digestion changed significantly under acute alkaline stress, and the variations from metabolic and immunity can cope with the osmotic shock of alkalinity stress in pacific white shrimp. This study provides key clues for exploring the molecular mechanism of pacific white shrimp under acute alkalinity stress, and also gives scientific basis for the optimisation of saline-alkaline aquaculture technology.

under accession numbers PRJNA1002106 and PRJNA1002632.

**Funding:** This research was funded by Gansu Natural Science Foundation (21JR7RA800). The funders had no role in study design, data collection and analysis, decision to publish, or preparation of the manuscript.

**Competing interests:** The authors have declared that no competing interests exist.

## Introduction

Nowadays, there are a lot of saline-alkali lands and secondary saline-alkali waters in the world, and they cannot be directly used for agriculture because of its high salinity and alkalinity, which have toxic effects on plants and animals [1]. The development of aquaculture in saline-alkaline lands can not only expand the aquaculture area but also reduce soil salinity and alkalinity, which has great significance to improve the ecology of saline-alkali soils [2]. Saline-alkaline fishery is one of the main inland aquaculture models developed in the past ten years [3]. With the maturity of aquaculture technologies, the saline-alkaline aquaculture area has expanded year by year, which has significant effects on ecological restoration and economic benefits [3]. At the same time, there are abundant saline-alkaline water resources in inland of Northwest China [3]. The main types of water alkalinity include carbonate, phosphate and borate alkalinity, while inland waters in Northwest China is mainly composed of carbonate system, with $CO_2$, $HCO_3^-$ and $CO_3^{2-}$ [4]. Although the high alkalinity of inland waters restrict the survival, growth, and reproduction of most aquatic animals, a part of species has been cultured in saline-alkali waters, including chinese mitten crab (*Eriocheir sinensis*), ridgetail white prawn (*Exopalaemon carinicauda*) and pacific white shrimp (*Litopenaeus vannamei*), etc [5].

The pacific white shrimp (*Litopenaeus vannamei*) is one of the three excellent species with the highest yield of shrimp culture in the world because of its fast growth, high meat yield and strong adaptability [6]. Because of its euryhaline characteristic ($0 \sim 50$ppt), pacific white shrimp have been successfully trialed in partial area of saline-alkali water, but no steady yield have been achieved [7, 8]. Its safe concentration of alkalinity for the juvenile is only 2.9 mmol/L, while the alkalinity of inland waters are even up to 44.5 mmol/L [9]. In addition, the alkalinity of inland waters always fluctuates considerably, leading to a series of stress reactions in pacific white shrimp, which is a key factor affecting the survival rate during shrimp rearing stage, and the acute stress reaction is obvious especially when the seedlings are placed in the pond [10]. Researchers preliminarily explored expression of genes related to saline-alkali tolerance by suppression subtractive hybridization and gene cloning [11, 12], but studies have shown that molecular response mechanism of alkalinity stress in crustaceans is a complex process controlled by the interaction of multiple genes [13]. It can be seen that present studies are limited on the effects of molecular response mechanism of alkalinity stress in crustaceans.

The homeostasis, digestion, metabolism and immunity of aquatic crustaceans have been greatly affected in adverse environment [14]. For example, the osmotic regulation, acid-base balance and ammonia transport of crustaceans mainly depends on enzymes of ion transport, such as $Na^+/K^+$-ATPase (NKA), carbonic anhydrase (CA), $Na^+/K^+/2Cl^-$ cotransporter (NKCC) and V-type ATPase (V-ATPase), to regulate the content of ions in the body. Through gene cloning and RNA interference, it was found that the gene expression of CA in ridgetail white prawn (*Exopalaemon carinicauda*) increased significantly under pH and saline–alkaline stresses, and after CA gene silencing, the gene expression of NKA also decreased [15]. Adverse environment inhibited the digestion and metabolism of crustaceans. It is found that the activities of protease, amylase and lipase in aquatic crustaceans change in adverse environment, such as salinity, pH, temperature, nutrient level and other environments [16]. Studies have shown that the digestive enzyme activity of redclaw crayfish (*Cherax quadricarinatus*) decreased with the increase of dietary protein [17]. In the process of metabolism, energy metabolism, as one of the ways for crustaceans to provide extra energy, can ensure osmotic regulation and ion exchange, such as lipid metabolism [18, 19]. Lipid metabolism includes fatty acid oxidation, cholesterol metabolism and phospholipid metabolism [20]. CPT1 (carnitine palmitoyl transferase 1) was the key genes of fatty acid oxidation [21–23]. For example, it was found that CPT I was involved in the *β*-oxidation of fatty acids, and freshwater shrimp

showed a high capacity for energy generation [24]. In terms of immunity, a series of immune-related enzymes, such as acid phosphatase (ACP), alkaline phosphatase (AKP), phenoloxidase (PO), superoxide dismutase (SOD), are important indicators to measure the immune function and health status of crustaceans [25–27]. In the experiment of studying the immune enzymes in ridgetail white prawn (*Exopalaemon carinicauda*) under carbonate alkalinity stress, the activities of ACP, AKP and SOD all increased within a short period of time [25]. Although there have been some studies on homeostasis, digestion, metabolism and immunity of crustaceans in adverse environment, the molecular response mechanism of crustaceans under alkaline stress by transcriptome sequencing is still limited.

Transcriptome sequencing can both analyze the genes expression and reveal its regulatory mechanism [28], and miRNAs usually bind to partially complementary 3' untranslated regions of target mRNA in aquatic animals and regulate the translation of target genes [29]. The recent studies can effectively reveal the expression patterns and regulation mode of genes under environmental stress through miRNA-mRNA analysis [30, 31]. Transcriptome sequencing and miRNA-mRNA targeting relationship analysis can interpret complex life processes and provide a powerful tool for interpreting the complex molecular response mechanism [32]. Therefore, this study screened candidate genes and related pathways for alkalinity stress through transcriptome sequencing and miRNA-mRNA combined analysis, and explored the expression profiles of miRNA and miRNA-mRNA related to alkalinity stress. This research provides an important theoretical basis for interpreting the molecular response mechanism of pacific white shrimp under alkaline environment, and also provides scientific basis for the optimisation of existing crustaceans aquaculture technology.

## Materials and methods

### Ethics statement

The animal study was reviewed and approved by Gansu Agricultural University (applicant number: GSAU-Eth-AST-2022-015). All experiments were carried out according to the "Chinese Laboratory Animal Care and Use Guide". The collected alive shrimps were placed on ice surface for low-temperature anesthesia during sampling, and the anesthetized shrimps were placed in centrifuge tubes that had been inserted into the ice. After the collection was completed, the centrifuge tubes were immediately moved into -80°C liquid nitrogen.

### Animals, stress treatments and sample collection

The juveniles of pacific white shrimp were obtained from Linfa Agriculture Company (Jiangsu, China). Desalted juveniles (length: 2.5±0.5 cm, weight: 0.40±0.5 g) were reared in aquaculture tanks containing basic water (12 h: 12 h dark / light cycle, salinity: 2‰, pH: 8.5, temperature: 27±0.5°C, alkalinity: 30 ∼ 50 mg/L, DO (dissolved oxygen): 7±1 mg/L, total hardness: 150±10 mg/L) with aerated tap-water, and temporarily cultured in Aquatic Science Training Center of Gansu Agricultural University (Gansu, China). Basic water (except $HCO_3^-$ and $CO_3^{2-}$) was prepared artificial seawater B solution [33], and the alkalinity was fine-tuned with 0.1 mol/L HCl and 0.1 mol/L NaOH. The alkalinity and pH were determined by acid-base titration every day [34]. One third of the water in each tank was renewed daily, and shrimps were fed standard diet at 10% of body weight twice a day.

Based on the studies related to the toxicity of carbonate alkalinity on pacific white shrimp [9, 11, 13], we set the preliminary experiment before formal experiment, and the semi-lethal alkalinity concentration ($LC_{50}$) of experimental shrimp in 24h was calculated (S1 Table). A control group (C, alkalinity of 50 mg/L) and a treatment group (T, alkalinity of 350 mg/L) were established in this experiment, and the determining methods of alkalinity and pH were

the same as that of basic water. The carbonate alkalinity water was prepared with sodium carbonate (analytical reagent: AR), sodium bicarbonate (AR) and basic water (acute carbonate alkalinity in the following articles is referred to as: alkalinity), and used after stability for 24 h. According to the results of the pre-experiment and the related research data [35], the stress treatment time was set to 24 h. During the stress experiment, six tanks were used for alkalinity stress experiment, including three groups of controls and three groups of high alkalinity treatments. The culturing situation in alkalinity stress experiment was the same as that during temporary culturing. Due to the difficulty of specific tissues obtaining, the juvenile samples in this size (2.5±0.5 cm) are generally collected in the form of whole shrimp [36], so the whole shrimp was also used as the research object in this study. At the time of sampling, five shrimps were collected from each tank and mixed into one sample, respectively. Three parallel test samples in per group, $C_{1\sim3}$ and $T_{1\sim3}$ were placed in 1.5 mL sterilized centrifuge tubes, frozen immediately in liquid nitrogen, and then stored at −80˚C. We set the mRNA sequencing group ($C_{m1\sim m3}$ and $T_{m1\sim m3}$) and the miRNA sequencing group ($C_{s1\sim s3}$ and $T_{s1\sim s3}$) in the process of data analysis.

### cDNA library construction and RNA sequencing

Total RNA was extracted using Trizol reagent kit (Invitrogen, Carlsbad, CA, USA), the RNAs concentration and quality were respectively assessed by an Agilent 2100 Bioanalyzer (Agilent Technologies, Palo Alto, CA, USA) and RNase free agarose gel electrophoresis. Eukaryotic mRNA was enriched by Oligo (dT) beads, while prokaryotic mRNA was enriched by removing rRNA by Ribo-ZeroTM Magnetic Kit (Epicentre, Madison, WI, USA). Then the enriched mRNA was fragmented into short fragments using fragmentation buffer and reverse transcripted into cDNA with random primers, followed by second-strand cDNA were synthesized by DNA polymerase I, RNase H, dNTP and buffer. The ligation products were size selected by agarose gel electrophoresis, PCR amplified, and sequenced using Illumina HiSeq2500 by Gene Denovo Biotechnology Co (Guangzhou, China).

For miRNA sequencing, a size range of 18∼30 nt RNA bands were isolated by polyacrylamide gel electrophoresis, and 5' and 3' adapters were ligated. The final PCR products were generated after reverse-transcription PCR and sequenced on the Illumina Novaseq6000 by Gene Denovo Biotechnology Co.

### DEGs and DEMs analysis and functional enrichment

RNAs differential expression analysis was performed by DESeq2 (v.1.6.3) software between C and T groups. The genes with the parameter of false discovery rate (FDR) < 0.05 and |Log2 fold change|≥2 were considered differentially expressed genes (DEGs). miRNAs differential expression analysis was performed by edgeR (v.3.6) software between C and T groups. We identified miRNAs with a fold change≥2 and $p < 0.05$ in a comparison as differentially expressed miRNAs (DEMs).

To assess the functional annotation and pathway enrichment, Gene Ontology (GO) and Kyoto Encyclopedia of Genes and Genomes (KEGG) analyses of DEGs and targets of DEMs were performed. The statistical enrichment of GO terms and KEGG pathway with $p < 0.05$ were regarded as statistically significant [37–39].

### Target genes prediction of DEMs and miRNA-mRNA regulatory network construction

The target genes of DEMs were predicted by RNAhybrid (v2.1.2) + svm_light (v6.01), Miranda (v3.3a), TargetScan (v7.0). The intersection of results were more credible to be chosen as

predicted miRNA target genes. miRNA sequences and family information were obtained from TargetScan website (http://www.targetscan.org/). The Pearson correlation coefficient (PCC) was used to define all possible miRNA-mRNA interactions, as well as the positive and negative relationships between miRNA and mRNA expression. For multi-group, pairs with PCC < -0.7 and $p < 0.05$ were selected as co-expressed negatively miRNA-mRNA pairs, and all RNAs were differentially expressed. The concerned negative relationship of miRNA-mRNA pairs were screened out to construct the regulatory network with Cytoscape software (v3.6.0) (http://www.cytoscape.org/).

### Verification of DEGs and DEMs by qRT-PCR

To validate the reliability of sequencing data, 13 genes with important stress functions were selected for qRT-PCR analysis. The total RNA used for the previous RNA-Seq and small RNA analysis was reverse transcribed into cDNA using a mixture of Stem loop primers and real-time primers respectively. qRT-PCR cycles were as follows: 30 s at 95°C, followed by 40 cycles of 5 s at 95°C and 30 s at 60°C. $\beta$-actin was used as internal controls to normalize the experimental data of the miRNAs and mRNAs. All primers were listed in S2 Table. mRNAs and miRNAs expression levels relative to the reference gene were calculated using the $2^{-\Delta\Delta CT}$ method. qRT-PCR data were presented as means±standard deviation (SD).

## Results

### Overview of mRNA-Seq and miRNA-Seq

To better understand the mechanism after stress in pacific white shrimp, we performed a comparative analysis of the mRNA ($C_m$ and $T_m$) and miRNA ($C_s$ and $T_s$) expression profiles of the C and T groups. A total of 299,686,068 ($C_m$) and 300,111,238 ($T_m$) raw reads were obtained from libraries of mRNA-seq. In order to ensure data quality, we filtered the original data before information analysis, and obtained 299,125,704 ($C_m$) and 299,552,630 ($T_m$) clean reads. Overviews of the specific sequencing and assembly results for $C_m$ and $T_m$ groups were shown in Table 1. The average quality of Q20 and Q30 in each library were higher than 98.22% and 94.78%, respectively, implying that the sequencing data were of high quality. In six libraries, the GC content ranged from 57.29% to 59.37%, and 85.28% to 88.61% of each library was mapped to the reference genome. The PCC of sample expression were greater than 0.943, suggesting that the reliability among these replicates and sequencing data produced in this study could be used for subsequent analysis (S1A Fig). Based on the comparison to reference genome sequence, 772 novel genes, 24,977 known genes and 25,749 total genes were identified in C and T groups, and the detailed comparison information of $C_1$, $C_2$, $C_3$, $T_1$, $T_2$ and $T_3$ were shown in the Fig 1A and S3 Table. $C_1$, $C_2$ and $C_3$ represented the three replicates of the control group, and $T_1$, $T_2$ and $T_3$ represented the three replicates of the treatment group.

A total of 35,130,519 ($C_s$) and 35,839,497 ($T_s$) raw reads were obtained from six miRNA libraries. After filtering the original reads, 34,817,492 and 35,544,755 high quality reads were obtained (Table 2). The total mapped tags to the reference genome in the $C_s$ and $T_s$ groups were 33,791,428 (96.17%) and 34,194,768 (95.42%), respectively. The length distribution of miRNAs in six libraries mainly concentrated in 21∼23 nt, of which 22 nt was the most abundant (S1B Fig). By comparing with the reference genome sequence, a total of 294 novel miRNAs, 335 known miRNAs and 629 total miRNAs were identified in C and T groups, and the detailed comparison information ($C_1$, $C_2$, $C_3$, $T_1$, $T_2$ and $T_3$) in the Fig 1B and S3 Table.

To confirm the mRNA and miRNA sequencing data, 7 DEGs and 6 DEMs were selected for qRT-PCR validation. As expected, the qRT-PCR data were in agreement with the results of

**Table 1. Statistical results of comparison rate between reads and reference genomes.**

| Sample name | C$_{m-1}$ | C$_{m-2}$ | C$_{m-3}$ | T$_{m-1}$ | T$_{m-2}$ | T$_{m-3}$ |
|---|---|---|---|---|---|---|
| Raw reads | 100,901,498 | 96,740,538 | 102,044,032 | 99,210,564 | 80,597,982 | 120,302,692 |
| Clean reads | 100,711,406 | 96,557,310 | 101,856,988 | 99,031,416 | 80,462,678 | 120,058,536 |
| Q20 (%) | 98.13% | 98.15% | 98.24% | 98.30% | 98.26% | 98.24% |
| Q30 (%) | 94.56% | 94.61% | 94.84% | 94.96% | 94.89% | 94.85% |
| GC content (%) | 57.96% | 57.29% | 58.27% | 58.69% | 59.33% | 59.37% |
| Total mapped | 38,746,463 (85.28%) | 40,171,923 (86.79%) | 39,976,627 (88.24%) | 36,845,744 (88.32%) | 27,957,008 (88.61%) | 41,134,936 (87.82%) |
| Multiple mapped | 6,186,611 (13.62%) | 7,327,739 (15.83%) | 7,261,977 (16.03%) | 6,243,380 (14.97%) | 5,433,020 (17.22%) | 7,561,987 (16.14%) |
| Unique mapped | 32,559,852 (71.66%) | 32,844,184 (70.96%) | 32,714,650 (72.21%) | 30,602,364 (73.36%) | 22,523,988 (71.39%) | 33,572,949 (71.67%) |

RNA-seq in terms of expression trends between C and T groups, which indicated the high reproducibility and reliability of the RNA-seq results in our study (Fig 2).

## Analysis of DEGs

By mRNA sequencing data analysis, the study detected 25,749 genes and assessed the distribution of gene expression abundance for each sample (S1C Fig). A total of 215 significant DEGs were identified in group T compared with group C, of which 180 DEGs were up-regulated and 35 DEGs were down-regulated (Fig 3A and S4 Table). The veen diagram indicated the overall number of differential genes with 353 DEGs were co-expressed (Fig 3B). Hierarchical clustering analysis of all DEGs showed high repeatability (Fig 3C). Among these DEGs, it was worth noting that many ion regulation, metabolism, digestion and immunity related DEGs were significantly up-regulated, and expression data of their expression and subvariety clustering between C and T groups were respectively shown in Fig 3D.

## GO and KEGG analysis of DEGs

To further investigate the function of DEGs, GO enrichment and KEGG pathway analyses were conducted. Hypergeometric tests were then applied to identify 49 GO terms that were significantly enriched in differential genes compared to background (Fig 4A and S3 Table). In the biological process categories single-organism process (GO:0044699), cellular process (GO:0009987), metabolic process (GO:0008152), response to stimulus (GO:0050896), biological regulation (GO:0065007) and regulation of biological process (GO:0050789) were

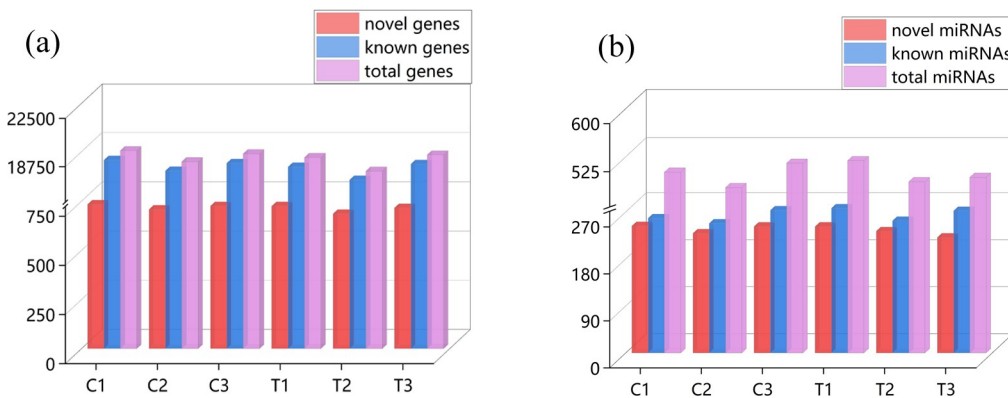

**Fig 1. Statistical analysis on the number of genes and miRNAs.** (a) The number of novel genes, known genes and total genes. (b) The number of novel miRNAs, known miRNAs and total miRNAs.

**Table 2. Categorization of pacific white shrimp non-coding and organellar small RNAs.**

| Sample name | $C_{s-1}$ | $C_{s-2}$ | $C_{s-3}$ | $T_{s-1}$ | $T_{s-2}$ | $T_{s-3}$ |
|---|---|---|---|---|---|---|
| total reads | 11,560,614 | 10,126,498 | 13,443,407 | 12,463,666 | 12,298,952 | 11,076,879 |
| High quality reads | 11,445,869 | 10,040,041 | 13,331,582 | 12,352,328 | 12,211,510 | 10,980,917 |
| Mapped reads | 11,034,517 (95.45%) | 9,762,154 (96.40%) | 12,994,757 (96.67%) | 11,915,164 (95.60%) | 11,671,416 (94.90%) | 10,608,188 (95.77%) |
| Exon | 21,572 | 20,611 | 24,917 | 25,186 | 26,241 | 17,182 |
| Known miRNA | 7,528,248 (68.22%) | 6,301,276 (64.55%) | 8,705,291 (66.99%) | 7,274,829 (61.06%) | 7,246,453 (62.09%) | 6,960,091 (65.61%) |
| Novel miRNA | 1,258,631 (11.41%) | 1,445,055 (14.80%) | 1,975,416 (15.20%) | 1,657,078 (13.91%) | 1,615,424 (13.84%) | 1,243,327 (11.72%) |
| rRNA | 646,547 (5.86%) | 587,098 (6.01%) | 729,535 (5.61%) | 890,379 (7.47%) | 949,419 (8.13%) | 788,251 (7.43%) |
| tRNA | 25,970 (0.24%) | 29,141 (0.30%) | 23,348 (0.18%) | 56,165 (0.47%) | 25,860 (0.22%) | 54,674 (0.52%) |
| snRNA | 663 (0.01%) | 497 (0.01%) | 692 (0.01%) | 768 (0.01%) | 739 (0.01%) | 772 (0.01%) |
| snoRNA | 5 (0.00%) | 18 (0.00%) | 22 (0.00%) | 38 (0.00%) | 37 (0.00%) | 37 (0.00%) |
| Others | 10361332 (93.90%) | 9145400 (93.68%) | 12241160 (94.20%) | 10967814 (92.05%) | 10695361 (91.64%) | 9764454 (92.05%) |

significantly clustered in the entries for DEGs; among cellular components, cell (GO:0005623) and cell part (GO:0044464) were the most common, with followed by organelle (GO:0043226); among molecular functional categories, a significant proportion of clusters were classified as binding (GO:0005488) and catalytic activity (GO:0003824).

KEGG pathway analysis identified the top 30 pathways that were significantly different under alkalinity stress (Fig 4B and S4 Table). Among these pathways, "Infectious diseases", "Cardiovascular diseases", "Circulatory system", "Immune system", "Endocrine system" and "Digestive system" were the most significantly enriched pathway subclasses. In addition, a number of important pathways were also significantly enriched, the specific information was as follows. "Proximal tubule bicarbonate reclamation (ko04964)", "MAPK signaling pathway (ko04010)" and "cGMP—PKG signaling pathway (ko04022)" were related to homeostatic

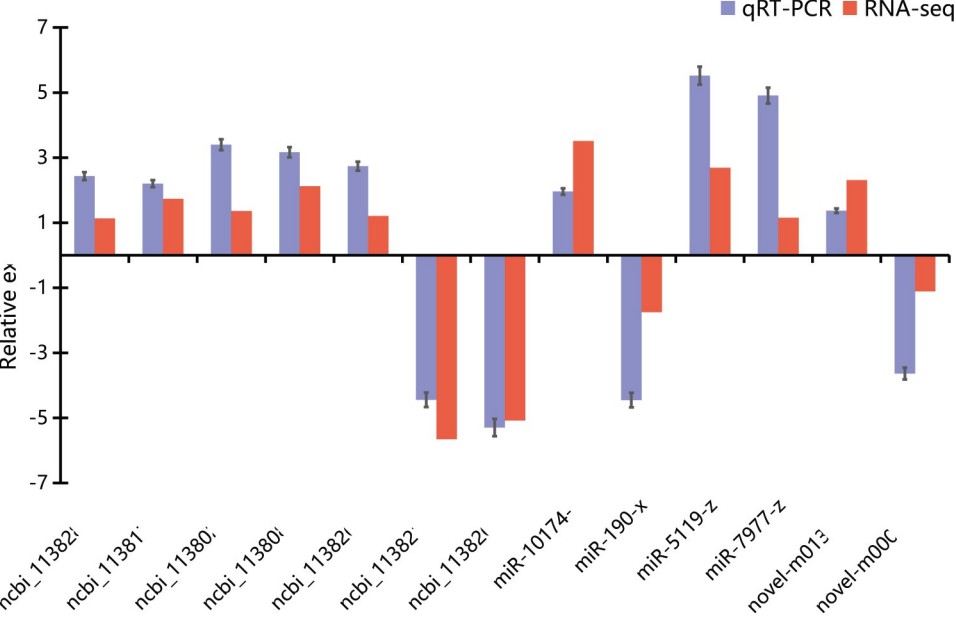

**Fig 2. Validation of DEGs and DEMs by qRT-PCR.** The expression comparison of DEGs and DEMs in RNA-seq and qRT-PCR. Error bars indicate standard deviation. Fold-change of qRT-PCR data represents the ratio of DEGs and DEMs expression values for T group comparing to C group after normalization against *β-actin*.

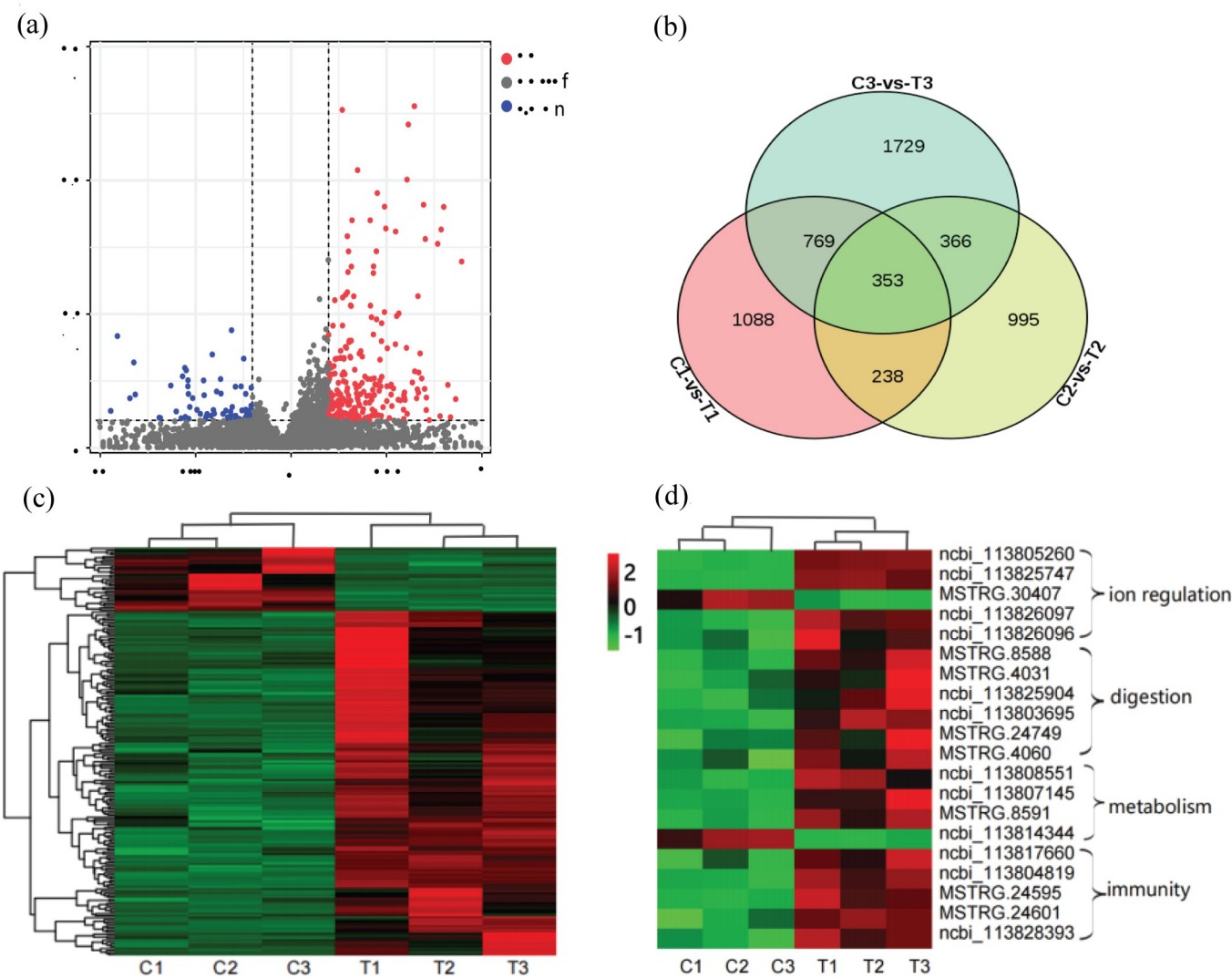

**Fig 3. Analysis of DEGs under alkalinity stress.** (a) Volcano diagram of DEGs; red dot means up-regulation, blue dot means down-regulation; (b) Venn diagram of $C_1$-vs-$T_1$, $C_2$-vs-$T_2$ and $C_3$-vs-$T_3$; (c) Heat map of DEGs; (d) Heat map of mRNA differential genes; showing differential genes related with ion regulation, digestion, metabolism and immunity.

regulation. "Carbohydrate digestion and absorption (ko04973)", "Salivary secretion (ko04970)" and "Pancreatic secretion (ko04972)" were related to digestion. The metabolism-related pathways were "Starch and sucrose metabolism (ko00500)" and "Metabolic pathways (ko01100)", and the immune-related genes enriched in pathways, including "Antigen processing and presentation (ko04612)" and "Glutathione metabolism (ko00480)".

## Analysis of DEMs

By mRNA sequencing data analysis, 35 significant DEMs were identified in T compared with C, of which 28 DEMs were up-regulated and 7 were down-regulated (Fig 5A). Details of up-regulated DEMs and down-regulated DEMs are shown in S5 Table. The overall number of differential genes detailed in Fig 5B. Also, we established heat map of some miRNAs related to ion regulation, metabolism, digestion and immunity (Fig 5C).

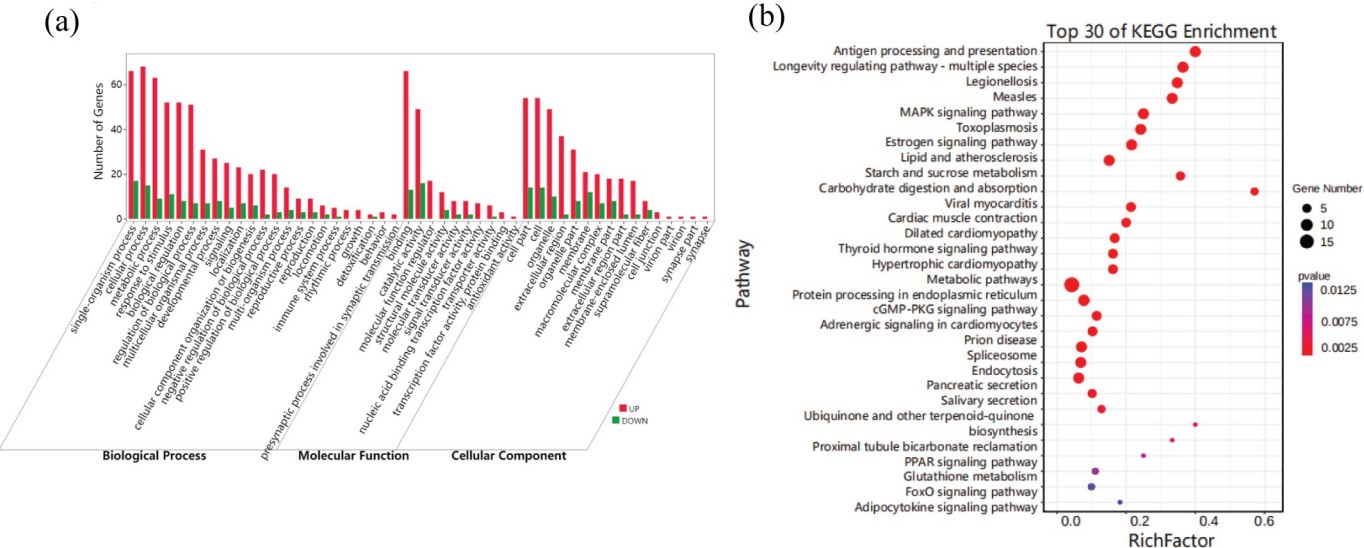

**Fig 4. Function analysis of DEGs under alkalinity stress.** (a) GO enrichment analysis of DEGs; red indicates up-regulation, green indicates down-regulation; (b) KEGG enrichment analysis of DEGs.

To understand the functional analysis of the predicted targets of miRNAs, we performed GO function and KEGG pathway analysis on target genes of DEMs. The GO results showed that the target genes of C and T were significantly enriched in differential genes with 66 GO terms (S1D Fig and S5 Table). Based on three categories, the 6 top terms were cellular process (GO:0009987), single-organism process (GO:0044699), response to stimulus (GO:0050896), cell (GO:0005623) and cell part (GO:0044464), which were generally consistent with the results of GO terms analysis of DEGs.

KEGG results showed that among these pathways, the most significantly enriched pathway class was "Human Diseases", and some important pathways were also significantly enriched, including "Environmental Information Processing", "Cellular Processes", "Genetic Information Processing", "Organismal Systems", and "Metabolism" subclass (Fig 5D and S5 Table). The "MAPK signaling pathway (ko04013)" related to homeostatic regulation; the "Lysosome (ko04142)" related to digestion; and the lipid-related genes were also enriched in the "Metabolic pathways (ko01100)" pathway.

## Construction of the miRNA-mRNA network

To explore the role of DEMs, we predicted the target genes that might have regulatory relationships with DEGs, and a total of 42 negatively associated miRNA-mRNA pairs were obtained, in which 35 DEMs and 215 DEGs were involved (S6 Table). Enrichment analysis of GO and KEGG were performed to filter the key terms and pathways, which played vital roles in the process of alkalinity stress with target genes of DEMs. We observed significant gene enrichment of GO terms that related to stress response including response to stimulus (GO:0050896), biological regulation (GO:0065007) and metabolic process (GO:0008152) (S7 Table). Additionally, KEGG enrichment analysis showed that many pathways were significantly enriched including "Glutathione metabolism (ko00480)", "Notch signaling pathway (ko04330)", "mTOR signaling pathway (ko04150)" and "Metabolic pathways (ko01100)" (S7 Table).

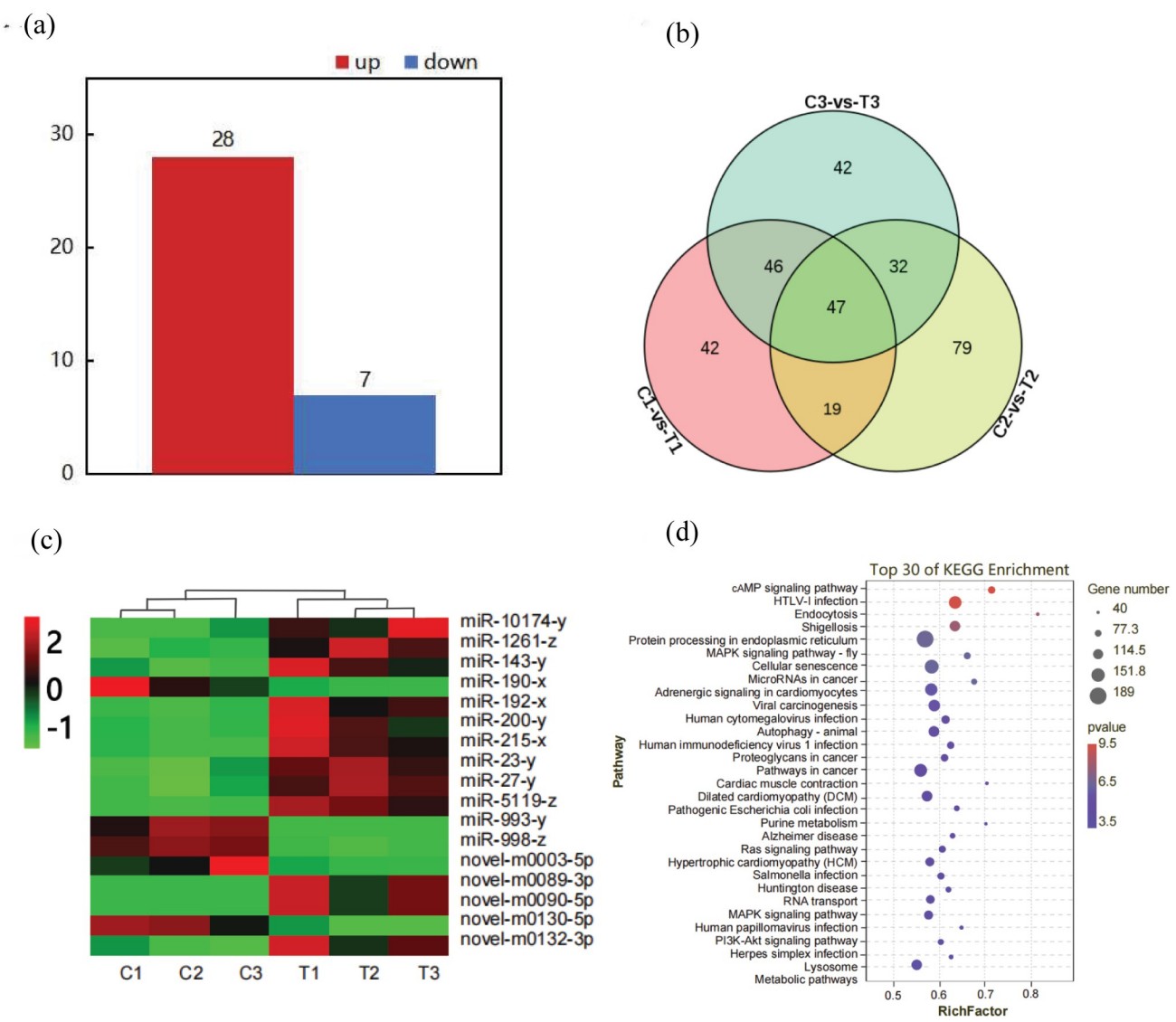

**Fig 5. Analysis of DEGs under alkalinity stress.** (a) The numbers represented the miRNAs up-regulated or down-regulated compared with the control; (b) Venn diagram of $C_1$-vs-$T_1$, $C_2$-vs-$T_2$ and $C_3$-vs-$T_3$; (c) Heat map of DEMs with significantly different expressions; (d) KEGG enrichment analysis of DEMs.

Furthermore, to investigate the potential role of regulatory relationships between these target genes, we used Cytoscape (v3.6.0) to construct miRNA-mRNA interaction network of selected pathways (Fig 6A). miR-200-y, miR-5119-z, miR-998-z, novel-m0089-3p, novel-m0090-5p, novel-m0130-5p and novel-m0132-3p were the miRNA targets that regulated more mRNAs. We also constructed gene-pathway interaction network (Fig 6B). mRNAs and miR-NAs were related to ion regulation, digestion, metabolism and immunity, and they main enriched in the pathways as below "Glutathione metabolism (ko00480)", "Carbohydrate digestion and absorption (ko04973)", "Salivary secretion (ko04970)", "Pancreatic secretion (ko04972)" and "Starch and sucrose metabolism (ko00500)".

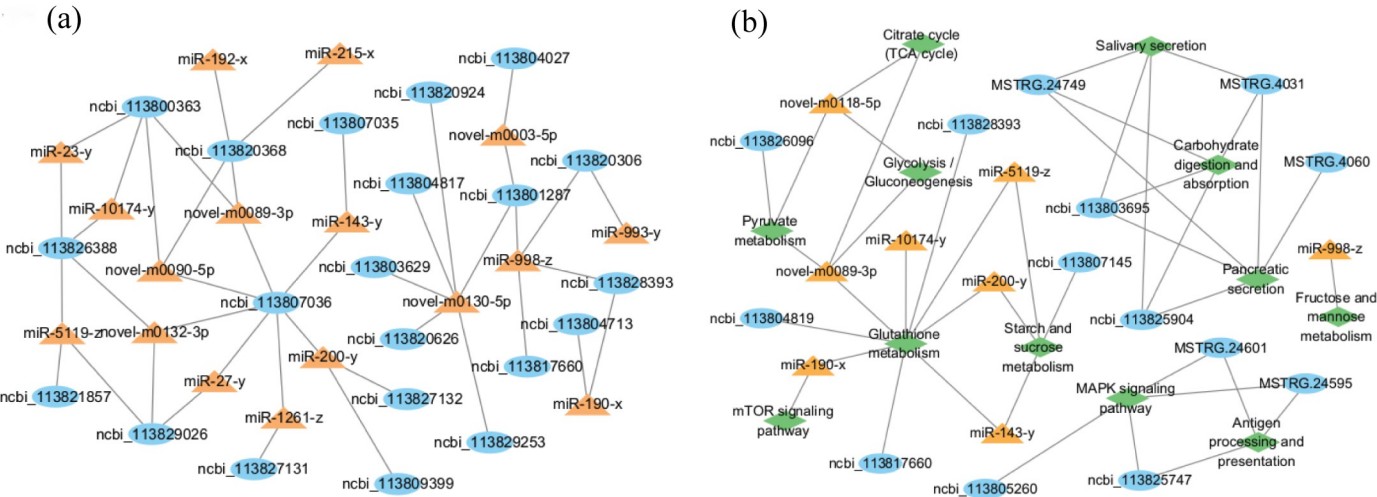

**Fig 6. Combined analysis of mRNA and miRNA under alkalinity stress.** (a) miRNAs and target mRNAs network analysis; (b) gene-pathway network. Blue circles indicate mRNAs, yellow triangles indicate miRNAs, and green diamonds indicate the enriched pathways.

## Discussion

In present study, we investigated the molecular response mechanism of acute alkalinity stress in juveniles of pacific white shrimp by transcriptome sequencing, and found that both mRNAs and miRNAs showed significant changes in homeostatic regulation, digestion, metabolism and immunity. These were of great significance for answering the question of molecular response mechanism under acute alkalinity stress and provided scientific basis for the optimisation of existing aquaculture technology. The specific information discussed is as follows.

### Effects of acute alkalinity stress on homeostatic regulation

Gill tissues of crustaceans play a key role in gas exchange, acid-base homeostasis and ion regulation [40]. Moreover, the gill is in close contact with the aqueous environment, which is often damaged and inflamed when there are drastic changes in environmental factors [41, 42], meanwhile accompanied dysfunction [43–46]. A series of ion transport-related enzymes and transporters in the gill play important roles in the regulation of homeostasis in crustaceans [47]. Na$^+$/K$^+$-ATPase (NKA), found in crustacean neurons [48], plays a vital role in maintaining cellular Na$^+$/K$^+$ homeostasis. NKA is evidenced as an ion transporter enzyme in the gills on fish [49], and it is a key enzyme for hypersalinity adaptation on blue crabs (*Callinectes sapidus*) [50]. The activity of NKA in gill epithelial cells increased following the change of salinity in aquatic crustaceans [51]. Carbonic anhydrase (CA) has essential functions in ion transport and acid-base homeostasis on crustaceans [52], and its activity is shown to be salinity sensitive in both high and low salinity in pacific white shrimp [53]. CA is widely distributed in crustacean tissues and functions in ion transport by providing H$^+$ and HCO$_3^-$, participating in the exchange transport of Na$^+$/H$^+$ and Cl$^-$/HCO3$^-$ [53]. Na$^+$/K$^+$/2Cl$^-$cotransporter (NKCC) has a conspicuous regulatory effect on osmotic pressure in crustaceans [54–58]. In swimming crab (*Portunus trituberculatus*) exposed to high salinity, mRNA expression of the Na/K/2Cl cotransporter was dramatically increased in the gills after 12 h [55]. Thus, NKA, CA and NKCC are essential indicators of the ability to study ion transport and acid-base regulation of crustaceans in stressful environment, so the analysis of these enzymes is more accurate for studying the regulatory capacity of the internal environment under alkalinity stress.

In this study, we found that the mRNA expressions of NKA (Na$^+$/K$^+$-ATPase subunit α: ncbi_113815921, ncbi_113815924; Na$^+$/K$^+$-ATPase subunit β: MSTRG.29808, ncbi_113810268, ncbi_113812436, ncbi_113827695), CA (carbonic anhydrase: ncbi_113803091, ncbi_113803454, ncbi_113830048, etc; carbonic anhydrase-related protein: ncbi_113802696, ncbi_113811760, ncbi_113812125, etc.) and NKCC (bumetanide-sensitive sodium-(potassium)-chloride cotransporter-like: ncbi_113830543, ncbi_113830547) were up-regulated. GO enrichment analysis of DEGs and DEMs revealed significant clustering of genes in response to stimulus terms with miR-200-y targeted regulation of ncbi_113809399, miRNA-5119-z targeted regulation of ncbi_113821857 were down-regulated, and these genes were also down-regulated in GO terms of combined analysis of mRNA and miRNA, indicating that miR-200-y positively regulated target gene ncbi_113809399, and miR-200-y contributed to further alleviation of alkalinity stress in pacific white shrimp through positive regulation of target gene. The clustering of genes in the terms "catalytic activity", "nucleic acid binding transcription factor activity" and "transporter activity" were found in the GO terms combined analysis of the study. In addition, KEGG enrichment analysis showed, the expression of ncbi_113805260, ncbi_113825747, ncbi_113826097 and ncbi_113826096 enriched in "Proximal tubule bicarbonate reclamation", "MAPK signaling pathway" and "cGMP-PKG signaling pathway" were up-regulated, while the expression of MSTRG.30407 was down-regulated, indicating that ion regulation was fluctuating due to alkalinity stress.

Recent studies have found that the gill tissues showed obvious inflammatory reactions under alkalinity stress in crustaceans, resulting in tissue damage, but enzyme activity experiments reveal that the activity of NKA, CA and NKCC are increased [59–61], which can cope with the shock of alkalinity stress. In this study, we got similar results as above, which provided support for our experiment. The above morphological and physiological studies had proved the molecular stress mechanism from the side under alkalinity stress in our study.

In brief, the result showed that genes and miRNA of homeostasis-related enzymes were significantly altered in pacific white shrimp under acute alkalinity stress, thereby regulating ion concentration and osmotic pressure to complete the regulation of internal environment.

## Effect of acute alkalinity stress on digestive and metabolic capacity

Digestive enzymes are a kind of special proteins that catalyze various biochemical reactions in crustaceans, and the increase of their activities are beneficial to promote the digestion and absorption of nutrients [62–64]. Trypsin and chymotrypsin are the primary alkaline proteolytic enzymes in the hepatopancreas of pacific white shrimp, which play an essential role in digestion [65]. α-amylase plays an essential role in the digestion of carbohydrates for example glycogen and starch [66, 67]. The study of differentially expressed proteins in Chinese shrimp (*Penaeus chinensis*) following hypoxic stress have confirmed that up-regulation of α-amylase, which causes an increase in fermentation substrates such as glucose to maintain ATP production [66]. Thus, these digestive enzymes such as trypsin, pancreatic rennet and α-amylase are more important indicators for analysing the digestive capacity of crustaceans, but the current research on digestive enzymes are relatively limited under alkalinity stress. Previous studies on acute salinity stress have shown that the crustaceans under non-isotonic environment, the stored energy is consumed to adapt to external salinity changes, and the farther away from the isotonic point, the more energy they consume for osmotic adjustment and the greater the impact on growth [68, 69]. In previous studies, it was shown that shrimp gills require additional energy (mostly from fatty acids) under salt stress to ensure osmotic regulation and ion exchange of pacific white shrimp [70]. CPT1 is a rate-limiting enzyme in fatty acid oxidation, and its activity affects the *β*-oxidation of fatty acids in the mitochondrial matrix and regulates

lipid metabolism [71]. It is evident that the energy metabolism of crustaceans often changes under environmental stress.

In the present study, the trypsin (MSTRG.10483, MSTRG.2371, ncbi_113804699 and ncbi_113806161), chymotrypsin (ncbi_113803370, ncbi_113803418, ncbi_113805736 and ncbi_113805737) and α-amylase (ncbi_113803694, ncbi_113803695, ncbi_113817723, ncbi_113825898 and ncbi_113825904) all showed the genes expression of digestive enzyme were increased. GO enrichment analysis revealed significant clustering of genes in "growth" and "developmental process" terms, with miR-143-y targeted regulation of ncbi_113807035, miR-192-x, miR-215-x, novel-m0089-3p and novel-m0090-5p targeted regulation of ncbi_113820368 were down-regulated. KEGG enrichment analysis showed that genes were enriched in "Carbohydrate digestion and absorption", "Salivary secretion" and "Pancreatic secretion" pathways. Genes enriched in these pathways, such as MSTRG.8588, MSTRG.4031, ncbi_113825904, ncbi_113803695, MSTRG.24749 and MSTRG.4060, were up-regulated.

A number of lipid metabolic pathways, including "fatty acid biosynthetic pathway", "arachidonic acid metabolic pathway", "glycerophospholipid metabolic pathway" and "sphingolipid metabolic pathway" were identified in osmotically stressed chinese mitten crab (*Eriocheir sinensis*) [72]. In this study, the above pathways were also found in pacific white shrimp under alkalinity stress. The fatty acid oxidation is a very complex physiological process, and the CPT enzyme system is mainly involved in the entry of long-chain fatty acids into mitochondria. In this study, we found that the gene expression of CPT (CPT1: ncbi_113810183, ncbi_113810184 and ncbi_113821199; CPT2: ncbi_113812201) were reduced, which indicating that fatty acid oxidation in pacific white shrimp was inhibited under alkalinity stress. In the GO enrichment analysis of miRNA-mRNA co-expression analysis, genes were clustered in metabolic process terms, and the expression levels of metabolism-related genes miRNAs (miR-10174-y, miR-190-x, miR-5119-z) also changed. After alkalinity stress, the expression of mRNAs and miRNAs related to lipid metabolism changed. Among them, the target gene ncbi_113826388 regulated by miR-10174-y was down-regulated in metabolic process terms, and the expression of ncbi_113804713 regulated by miR-190-x was up-regulated, indicating that miRNAs and mRNAs had a targeting relationship under alkalinity stress of pacific white shrimp, which might be an important factor in regulating lipid metabolism of pacific white shrimp. KEGG enrichment analysis of DEGs showed that the genes were enriched in "Starch and sucrose metabolism" and "Metabolic pathways". In KEGG enrichment analysis of DEMs, the genes were enriched in "Metabolic pathways", and the genes such as ncbi_113808551, ncbi_113807145, MSTRG.8591 and ncbi_113814344 in the pathway were up-regulated.

In short, the result showed that genes and miRNA of digestive and metabolic related enzymes were significantly altered in pacific white shrimp under acute alkalinity stress, indicating that pacific white shrimp needed to cope with the energy shortage problem during acute alkalinity stress, and provided sufficient energy through digestion to maintain ion balance to reduce osmotic shock.

### Effects of acute alkalinity stress on immune function

Invertebrates lack acquired immune system, and non-specific immunity plays significant role in immunity and pathogen preventive control of invertebrates [73]. Glutathione is a sulfhydryl-rich active substance in organisms, which has vital biological functions such as anti-oxidation, detoxification and growth promotion [74, 75]. Glutathione has two forms: reduced (GSH) and oxidized (GSSG), the glutathione peroxidase (GPX) is ubiquitous in organisms, which catalyzes the reaction of GSH into GSSG, and the glutathione reductase (GR) is an important antioxidant enzyme that uses NADPH enzyme to catalyze the reaction of GSSG

into GSH, thereby providing reducing power for the intracellular scavenging of reactive oxygen species (ROS) and mitigating environmental damage to the organism [76]. Studies have shown that the activity of phenoloxidase (PO) in pacific white shrimp has altered following salinity mutations [77]. Alkaline phosphatase (AKP) and acid phosphatase (ACP) are essential indicators to measure the body's immune ability and health status, and are important factors in acid-base regulation of crustaceans [78]. Under external stress conditions, some metabolism in shrimp is regulated by phosphorylation and dephosphorylation of substances, these processes are catalyzed by different phosphatases, with ACP and AKP are directly involved in the transfer of phosphate groups [79, 80]. When stimulated by the external environment, crustaceans will cause stress response, and immune-related enzyme activity will increase within a certain range, which is a protective mechanism produced by the body, so it is essential to study the immune response of crustaceans under alkalinity stress.

In this study, we found that at the transcriptional level, after acute alkalinity stress, the expression of GPX (selenium-dependent glutathione peroxidase: MSTRG.10388 and MSTRG.12294), GPX-like: ncbi-113816684, ncbi_113826359 and ncbi-113800343), GSH (MSTRG.7471, ncbi_113829084, ncbi_113821223 and ncbi_113829262), PO (MSTRG.21794, ncbi_113824294 and ncbi_113802496), ACP (MSTRG.6311, ncbi_113818646, ncbi_113827042) and AKP (salivary alkaline phosphatase: MSTRG.16669 and MSTRG.16670) in pacific white shrimp were increased. GO enrichment analysis of DEGs and DEMs showed that the genes were significantly enriched in "detoxification", "immune system process" and "antioxidant activity", and the genes were also enriched in "detoxification" and "immune system process" in GO enrichment analysis of miRNA-mRNA co-expression analysis. The mRNA KEGG enrichment analysis showed that the genes were enriched in two pathways, including "Antigen processing and presentation" and "Glutathione metabolism", and the immune-related genes (ncbi_113817660, ncbi_113804819, MSTRG.24595, MSTRG.24601, and ncbi_113828393) were up-regulated. The KEGG enrichment analysis showed that genes (ncbi_113817660 and ncbi_113828393) were significantly enriched in "glutathione metabolism" (ko00480). The ncbi_113821857 gene regulated by miR-5119-z was up-regulated, and the ncbi_113828393 regulated by miR-998-z was down-regulated, indicating that miR-5119-z and miR-998-z were involved in the alkalinity stress process. The heat shock proteins target damaged proteins to the proteasome to prevent the accumulation of dysfunctional proteins and to cycle peptides and amino acids [81–83], which proved that high levels of autophagy may exist under alkalinity stress. In our study, heat shock 70-kDa protein was rapidly up-regulated in alkalinity stress, indicating that it played an essential role in alkalinity stress response. In summary, the result showed that genes and miRNA of immunity related enzymes were significantly altered in pacific white shrimp under acute alkalinity stress thereby exerting immune function.

## Conclusions

This study investigated the molecular response mechanism of pacific white shrimp under alkalinity stress by RNA-seq. Under acute alkalinity stress, the expression levels of most key alkalinity stress-related genes enriched in ion regulation, digestion and immunity increased, and the expression levels of genes enriched in lipid metabolism were down-regulated. This research indicated that the genes and miRNAs related to homeostatic regulation, digestive, metabolic and immunity have changed significantly under alkalinity stress in pacific white shrimp. The results provide basic data for further analyzing the molecular mechanism under alkalinity stress, and also provide theoretical basis for optimizing the culture technology in pacific white shrimp.

## Supporting information

**S1 Fig. Information figure of miRNA and mRNA.**
(TIF)

**S1 Table. Mortality of pacific white shrimp in 24 h under different carbonate alkalinity.**
(XLSX)

**S2 Table. The sequence of primers used in this study.**
(XLSX)

**S3 Table. Details of available mRNA number data.**
(XLSX)

**S4 Table. Differential mRNA details table; GO terms of DEGs; KEGG enrichment of DEGs.**
(XLSX)

**S5 Table. Differential miRNA details table; GO terms of DEMs; KEGG enrichment of DEMs.**
(XLSX)

**S6 Table. The miRNA-targeting mRNA genes.**
(XLSX)

**S7 Table. GO terms of miRNA-mRNA; KEGG enrichment of miRNA-mRNA.**
(XLSX)

## Acknowledgments

We thank the members of Gansu agriculture University for valuable discussions.

## Author Contributions

**Data curation:** Ruiqi Zhang.

**Methodology:** Xiang Shi.

**Supervision:** Zhe Liu, Jun Sun, Lanlan Li, Guiyan Zhao, Junhao Lu.

**Writing – original draft:** Xiang Shi.

**Writing – review & editing:** Ruiqi Zhang.

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
