## [Decision Letter · Decision Letter 0]

5 Jun 2023

PONE-D-23-12457Combined analysis of mRNA and miRNA reveals the mechanism of pacific white shrimp (Litopenaeus vannamei) under acute alkalinity stressPLOS ONE

Dear Dr. Zhang,

Thank you for submitting your manuscript to PLOS ONE. After careful consideration, we feel that it has merit but does not fully meet PLOS ONE’s publication criteria as it currently stands. Therefore, we invite you to submit a revised version of the manuscript that addresses the points raised during the review process.

We look forward to receiving your revised manuscript.

Kind regards,

Mahmoud A.O. Dawood, PhD

Academic Editor

PLOS ONE

3. Please make sure that all information entered in the 'Ethics Statement' section regarding ethics approval is also included in the Methods section of the manuscript

4. We suggest you thoroughly copyedit your manuscript for language usage, spelling, and grammar. If you do not know anyone who can help you do this, you may wish to consider employing a professional scientific editing service.

A clean copy of the edited manuscript (uploaded as the new *manuscript* file).

Reviewers' comments:

Reviewer's Responses to Questions

**Comments to the Author**

1. Is the manuscript technically sound, and do the data support the conclusions?

Reviewer #1: Yes

Reviewer #2: Yes

2. Has the statistical analysis been performed appropriately and rigorously? 

Reviewer #1: Yes

Reviewer #2: Yes

3. Have the authors made all data underlying the findings in their manuscript fully available?

Reviewer #1: Yes

Reviewer #2: Yes

4. Is the manuscript presented in an intelligible fashion and written in standard English?

Reviewer #1: Yes

Reviewer #2: Yes

5. Review Comments to the Author

Reviewer #1: The study by Shi et al explored mRNA and miRNA on the mechanism of pacific white shrimp (Litopenaeus vannamei) under acute alkalinity stress, the data were properly analyzed, and has certain significance for the saline-alkaline aquaculture of pacific white shrimp. The manuscript can be published, but there are concerns that need to be addressed.

1.Authors should provide the reason(s) for selection of the high alkalinity (350 mg/L) considered in this study.

2.When sampling, how many shrimps were mixed into one sample? Kindly clarify.

3.Lines 113 reads “Three parallel test samples in per group”, authors should describe the sampling situation, specifically authors should describe the situation of the three parallel samples.

4.Available phenotypic data such as body size of the selected juvenile should be provided.

5.Author should provide the cultivating conditions in the experimental process as much as possible.

6.Lines 179-180 “information of C1, C2, C3, T1, T2 and T3” should be written in detail. Like line 188.

7.The data available of the transcriptome sequencing used in this study should be included.

8.The title of Fig 1 is not appropriate consider “Statistics on the number of genes.

9.Lines 353-354 “During the long-term evolutionary…structure levels. ” must be referenced.

10.There is need to improve the grammar of this manuscript.

Reviewer #2: A combined analysis of mRNA and miRNA was done to reveal the mechanism of pacific white shrimp (Litopenaeus vannamei) under acute alkalinity stress. The manuscript was well organized, it could be accepted after some issues being solved.

1. The DEMs should be defined in the abstract.

2. The introduction should be re-written, especially the third paragraph.

3. How many shrimp in one sample?

4. It is better to detect the activity of digestive enzymes.

5. The formulate of the references should checked.

6. PLOS authors have the option to publish the peer review history of their article (what does this mean?). If published, this will include your full peer review and any attached files.

Reviewer #1: No

Reviewer #2: No

---

## [Author Response · Author response to Decision Letter 0]

17 Jul 2023

June. 6, 2023

Editor-in-Chief

PLOS ONE

Manuscript ID: PONE-D-23-12457

Title: Combined analysis of mRNA and miRNA reveals the mechanism of pacific white shrimp (Litopenaeus vannamei) under acute alkalinity stress

Dear editor:

We thank you and reviewers for reviewing our manuscripts and providing valuable comments. We have revised the manuscript carefully according to the reviewers’ suggestions and comments. Please find our enclosed responses to the reviewers’ comments. We hope the revised manuscript is suitable for publication in PLOS ONE.

Regards. 

Ruiqi Zhang

College of Animal Science & Technology, Gansu Agricultural University, Lanzhou

730070, China

E-mail: mading000000@126.com

Response to reviewers:

Reviewer #1: The study by Shi et al explored mRNA and miRNA on the mechanism of pacific white shrimp (Litopenaeus vannamei) under acute alkalinity stress, the data were properly analyzed, and has certain significance for the saline-alkaline aquaculture of pacific white shrimp. The manuscript can be published, but there are concerns that need to be addressed.

We are very grateful to your comments for the manuscript. According with your advice, we amended the relevant part in manuscript. All of your questions were answered below. At the same time, we made a comprehensive improvement to the language of the article.

1.Authors should provide the reason(s) for selection of the high alkalinity (350 mg/L) considered in this study.

Response: Based on the studies related to the toxicity of carbonate alkalinity on pacific white shrimp [1-3], we set the preliminary experiment before formal experiment, and the semi-lethal alkalinity concentration (LC50) of experimental shrimp in 24 h was calculated (Tab. S1). We found the 24 h mortality at 300 mg/L and 400 mg/L were 44.08 % and 56.23 %, respectively, while the semi-lethal concentration was between 300 mg/L and 400 mg/L, so we set the high alkalinity concentration to 350 mg/L. Specific references can be made to the following report:

Reference: [1] Yao Z L, Wang H, Zhou K, Ying C Q, Lai Q F. Effect of carbonate alkalinity and pH on the survival of shrimp. Journal of Ecology. 2010;29(5):945-950. doi:10.13292/j. 1000-4890.2010.0160.

Reference: [2] Yang F Y, Sun L M, Yang X Q. Toxic effects of carbonate alkalinity on juvenile of pacific white shrimp. Aquatic Science. 2004; 9: 3-6. https://doi.org/10.1016/j.etap.20 14.08.006

Reference: [3] Yang F Y, Li X J, Yang X Q, Sun L M. Adaptation of pacific white shrimp to alkaline waters. Agricultural Bulletin of China. 2005; 8: 413-416. https://doi.org/10.1 007/s118 02-022-4882-9 

The relevant descriptions have been marked in the manuscript as follows (Lines 118-121). We hope that the revised manuscript meets your requirements.

“... Based on the studies related to the toxicity of carbonate alkalinity on pacific white shrimp, we set the preliminary experiment before formal experiment, and the semi-lethal alkalinity concentration (LC50) of experimental shrimp in 24h was calculated (Tab. S1)...”

2.When sampling, how many shrimps were mixed into one sample? Kindly clarify.

Response: During sampling, in order to meet the content of RNA that can be extracted from a sample, we collected five shrimps and put them in a centrifuge tube as a sample. The related descriptions have made detailed modifications in our manuscript, and the specific modifications are as follows (Lines 133-134): 

“... At the time of sampling, five shrimps were collected from each tank and mixed into one sample, respectively...” 

3. Lines 113 reads “Three parallel test samples in per group”, authors should describe the sampling situation, specifically authors should describe the situation of the three parallel samples.

Response: Thank you for your comment. In this study, we set up 6 tanks, three of which were used as control groups, and the other three aquaculture tanks were used as treatment groups of high alkalinity. The culturing situation in alkalinity stress experiment was the same as that during temporary culturing. At the time of sampling, five shrimps were collected from each tank and mixed into one sample, respectively. The changes of the relevant description in the manuscript are as follows (Lines 127-130). 

“... During the stress experiment, six tanks were used for alkalinity stress experiment, including three groups of controls and three groups of high alkalinity treatments. The culturing situation in alkalinity stress experiment was the same as that during temporary culturing...”

4. Available phenotypic data such as body size of the selected juvenile should be provided. 

Response: Thank you for your comment. In this study, we selected the juvenile shrimps (length: 2.5±0.5 cm, weight: 0.40±0.5 g) for temporary culture and stress experiments. Available phenotypic data has been added in the article. Details are as follows (Line 109 and Line 131).

“... Desalted juveniles (length: 2.5±0.5 cm, weight: 0.40±0.5 g) were reared in aquaculture tanks...”

5.Author should provide the cultivating conditions in the experimental process as much as possible.

Response: Thank you for your comment. Details of the cultivating conditions are as follows. For example, 12 h : 12 h dark / light cycle, salinity: 2‰, pH: 8.5, temperature: 27±0.5°C, alkalinity: 30∼50 mg/L, DO (dissolved oxygen): 7±1 mg/L, total hardness: 150±10 mg/L.

The related descriptions have been changed in the manuscript, and the particulars are as follows (Lines 110-112). We hope that the revised manuscript meets your requirements.

“... Desalted juveniles (length: 2.5±0.5 cm, weight: 0.40±0.5 g) were reared in aquaculture tanks containing basic water (12 h : 12 h dark / light cycle, salinity: 2‰, pH: 8.5, temperature: 27±0.5°C, alkalinity: 30∼50 mg/L, DO (dissolved oxygen): 7±1 mg/L, total hardness: 150±10 mg/L) with aerated tap-water, and temporarily cultured in Aquatic Science Training Center of Gansu Agricultural University...”

6.Lines 179-180 “information of C1, C2, C3, T1, T2 and T3” should be written in detail. Like line 188. 

Response: Thank you for your comment. A control group (C, alkalinity of 50 mg/L) and a treatment group (T, alkalinity of 350 mg/L) were set in this experiment. C1, C2 and C3 represent the three replicates of the control group, and T1, T2 and T3 represent the three replicates of the treatment group. Based on the comparison to reference genome sequence, 772 novel genes, 24,977 known genes and 25,749 total genes were identified in C and T groups. Available mRNA number data of C1, C2, C3, T1, T2 and T3 have been added to Table S3 with details below.

S3 Table. Details of available mRNA number data

 novel genes known genes total genes

C1 734 18097 18831

C2 708 17261 17969

C3 725 17856 18581

T1 725 17572 18299

T2 686 16549 17235

T3 716 17784 18500

By comparing with the reference genome sequence, a total of 294 novel miRNAs, 335 known miRNAs and 629 total miRNAs were identified in C and T groups. Available miRNA number data of C1, C2, C3, T1, T2 and T3 have been added to Table S3 with details below.

S3 Table. Details of available miRNA number data

 novel miRNAs known miRNAs total miRNAs

C1 243 258 501

C2 229 248 477

C3 242 273 515

T1 242 277 519

T2 233 253 486

T3 221 272 493

The related descriptions have been added in the manuscript, and the particulars are as follows (Lines 206-208, and Line 217). 

“... And the detailed comparison information of C1, C2, C3, T1, T2 and T3 were shown in the Fig 1a and S3 Table. C1, C2 and C3 represented the three replicates of the control group, and T1, T2 and T3 represented the three replicates of the treatment group...”

7.The data available of the transcriptome sequencing used in this study should be included.

Response: The datasets presented in this study have been supplemented in the manuscript, and the details are as follows (Lines 140-142). 

“ The datasets presented in this study can be found in online repositories. The names of the repository/repositories and accession number(s) can be found below: NCBI under accession number GSE235873...”

8.The title of Fig 1 is not appropriate consider “Statistics on the number of genes.

Response: Thank you for your suggestion. The necessary changes have been made in the manuscript, and the details are as follows (Line 224).

“ Fig 1. Statistical analysis on the number of genes and miRNAs.”

9.Lines 353-354 “During the long-term evolutionary…structure levels. ” must be referenced.

Response: Thank you for your comment. The references have been added, and some of the expressions were not clear enough in this section. Now we have made detailed modifications, and the specific modifications are as follows: (Lines 383-385). 

“Digestive enzymes are a kind of special proteins that catalyze various biochemical reactions in crustaceans, and the increase of their activities are beneficial to promote the digestion and absorption of nutrients...”

10.There is need to improve the grammar of this manuscript.

Response: Thank you for your comment. we made a comprehensive improvement to the language of the article. 

Reviewer #2: A combined analysis of mRNA and miRNA was done to reveal the mechanism of pacific white shrimp (Litopenaeus vannamei) under acute alkalinity stress. The manuscript was well organized, it could be accepted after some issues being solved.

We are very grateful to your comments for the manuscript. According with your advice, we amended the relevant part in manuscript. All of your questions were answered below. At the same time, we made a comprehensive improvement to the language of the article. 

1.The DEMs should be defined in the abstract.

Response: Thank you for your valuable comment. The necessary changes have been made in the abstract, and specific modifications are as follows (Lines 16).

“... We identified 215 differentially expressed mRNAs (DEGs) and 35 differentially expressed miRNAs (DEMs)...”

2.The introduction should be re-written, especially the third paragraph.

Response: Thank you for your comment. The detailed modifications in the introduction section have been made based on the research background and objectives to highlight the purpose and significance of this study. The specific modifications have been highlighted in red for your review. (Pages 3-5).

3.How many shrimp in one sample?

Response: During sampling, in order to meet the content of RNA that can be extracted from a sample, we collected five shrimps and put them in a centrifuge tube as a sample. The necessary additions have been made in our manuscript, and the specific modifications are as follows (Lines 133-134): 

“... At the time of sampling, five shrimps were collected from each tank and mixed into one sample, respectively...” 

4.It is better to detect the activity of digestive enzymes.

Response: Thank you for your review and suggestions. We fully agree with your revision of this part. As an analysis of this part, it is necessary to use enzyme activity for further verification. We had determined the enzyme activity in related study and found that digestive enzyme activity changed significantly under acute alkalinity stress, which is consistent with the content of this study. At the same time, because this part of the data has been published in related articles, we do not add the data of enzyme activity in this study. What’s more, in view of the fact that we did not explain this in the previous study and might be confused when reading the article, so we made appropriate modification in the article, and cited the report of our study. We hope that the relevant modifications of this part can meet your requirements.

Reference: [4] Zhang R Q, Shi X, Liu Z, Sun J, Sun T Z, Lei M Q. Histological, Physiological and Transcriptomic Analysis Reveal the Acute Alkalinity Stress of the Gill and Hepatopancreas of Litopenaeus vannamei. Marine biotechnology. 2023; 1-15. https://doi:10.1007/ S10126- 023-10 228-1.

5.The formulate of the references should checked.

Response: Thank you for your comment. In response to this problems, we have made substantial amendments to the references during the revision period.

---

## [Decision Letter · Decision Letter 1]

3 Aug 2023

Combined analysis of mRNA and miRNA reveals the mechanism of pacific white shrimp (Litopenaeus vannamei) under acute alkalinity stress

PONE-D-23-12457R1

Dear Dr. Zhang,

We’re pleased to inform you that your manuscript has been judged scientifically suitable for publication and will be formally accepted for publication once it meets all outstanding technical requirements.

Kind regards,

Mahmoud A.O. Dawood, PhD

Academic Editor

PLOS ONE

Additional Editor Comments (optional):

Reviewers' comments:

Reviewer's Responses to Questions

**Comments to the Author**

1. If the authors have adequately addressed your comments raised in a previous round of review and you feel that this manuscript is now acceptable for publication, you may indicate that here to bypass the “Comments to the Author” section, enter your conflict of interest statement in the “Confidential to Editor” section, and submit your "Accept" recommendation.

Reviewer #1: All comments have been addressed

Reviewer #2: All comments have been addressed

2. Is the manuscript technically sound, and do the data support the conclusions?

Reviewer #1: Yes

Reviewer #2: Yes

3. Has the statistical analysis been performed appropriately and rigorously? 

Reviewer #1: Yes

Reviewer #2: Yes

4. Have the authors made all data underlying the findings in their manuscript fully available?

Reviewer #1: Yes

Reviewer #2: Yes

5. Is the manuscript presented in an intelligible fashion and written in standard English?

Reviewer #1: Yes

Reviewer #2: Yes

6. Review Comments to the Author

Reviewer #1: The author answered my comments and made revisions so that the manuscript could be published. I don't have extra attention.

Reviewer #2: (No Response)

7. PLOS authors have the option to publish the peer review history of their article (what does this mean?). If published, this will include your full peer review and any attached files.

Reviewer #1: No

Reviewer #2: No

---

## [Editor Report · Acceptance letter]

8 Aug 2023

PONE-D-23-12457R1 

Combined analysis of mRNA and miRNA reveals the mechanism of pacific white shrimp (Litopenaeus vannamei) under acute alkalinity stress 

Dear Dr. Zhang:

I'm pleased to inform you that your manuscript has been deemed suitable for publication in PLOS ONE. Congratulations! Your manuscript is now with our production department. 

Kind regards, 

on behalf of

Dr. Mahmoud A.O. Dawood 

Academic Editor

PLOS ONE